# Gaseous Mercury Limit Values: Definitions, Derivation, and the Issues Related to Their Application

Francesco Ciani [1,*], Pilario Costagliola [1], Pierfranco Lattanzi [2] and Valentina Rimondi [1]

1   Department of Earth Science, University of Florence, 50121 Florence, Italy; pilario.costagliola@unifi.it (P.C.); valentina.rimondi@unifi.it (V.R.)
2   Consiglio Nazionale delle Ricerche (CNR), Institute of Geosciences and Georesources, Unità di Firenze, 50121 Florence, Italy; pierfrancolattanzi@gmail.com
*   Correspondence: francesco.ciani@unifi.it; Tel.: +39-055-275-7506

**Abstract:** Mercury (Hg) is a toxic and persistent element, easily bio-accumulable in the food chain with several dangerous effects on people's health. Among Hg airborne species, gaseous elemental mercury (GEM) is dominant, more persistent in the atmosphere, and highly absorbable by humans. The issue of atmospheric Hg pollution is largely discussed by several environmental agencies, giving rise to a number of remarkably different threshold values beyond which exposure to Hg in its different species is deemed dangerous. The present paper presents a comprehensive compilation of the threshold limit values (TLVs) suggested/recommended by environmental health agencies regarding the exposition to airborne Hg. The work tries to clarify the applicability of the threshold concentrations, their terminology, and the methods by which they were calculated. The most critical key-points in Hg TLVs derivation and use are stressed. The literature revision identifies about 20 TLVs: among these, only four are legally transposed into environmental laws, while the majority are just recommendations. There is a high variability of suggested values for gaseous Hg TLVs, mostly resulting from the different methodologies applied for their calculation. This difference is the consequence of a considerable independence among agencies that suggest or enforce Hg limit values. However, in the past years, a generalized substantial lowering of the Hg TLVs, both for chronic and occupational exposure, has been observed. This tendency reflects a revision trend towards a more protective approach for people's health.

**Keywords:** mercury; threshold limit values; environmental laws; sustainability

## 1. Introduction

The perception of the scientific community on the risk related to mercury (Hg) exposure was originally reported in the mid-1960s, soon after the first outbreak of methyl-Hg poisoning of people living in Minamata Bay, Japan [1,2]. Only after more than forty years, the question was tackled on a worldwide scale with the release of the first edition of the Global Mercury Assessment (2002) by the United Nations Environment Program (UNEP, [3]). The first regulations to limit Hg release and use started soon after the UNEP assessment. Despite this, anthropogenic emissions overpassed two million kg of Hg in 2018, about 30% of the total Hg entering in the atmosphere [4].

The issue of atmospheric Hg pollution is now widely debated by the scientific community due to the centrality of the problem. Gaseous Hg is in fact a global pollutant, that can be transported thousands of kilometers in the atmosphere, biogenically transformed into more toxics forms and thus biomagnified in the food chain [5].

With respect to other heavy metals, Hg has a low but measurable vapor tension. Since Hg-containing items (thermometers, batteries, lamps etc.) are diffused in many environments including homes [6,7], humans could be unaware of the exposure to anomalous atmospheric $Hg^0$ concentrations which could be rather common and unavoidable. The

question is actively discussed by several international agencies, giving rise to various Hg limit values, mainly focused on the occupational framework. The large number of stakeholders involved in the question and the continuous scientific updates have produced a multitude of different limit values that apply to similar and often overlapping contexts. Terminology therefore is not always easy to understand.

The present paper offers an overview of the threshold limit values (TLVs) suggested/recommended by the government or health agencies regarding the exposition to gaseous Hg. Apart from international agencies such as WHO, the present study is limited to Western countries, including European Union, USA, Canada, Australia, and New Zealand. The work tries to clarify the applicability of the threshold concentrations, the terminology, and the methods of derivation. The most critical key points in Hg TLVs derivation and use will be stressed.

## 2. Mercury Forms, Sources and Health Effects

Mercury is one of the most toxic elements naturally present on the Earth's crust; it is easily bio-accumulable in the food chain, with highly dangerous effects on human's health [8].

Airborne Hg species are divided among gaseous species, i.e., gaseous elemental Hg (GEM or $Hg^0$) and reactive gaseous Hg (RGM, $Hg^{2+}$), and particulate species, i.e., particulate-bound Hg (PBM). The sum of GEM and RGM is often defined as total gaseous Hg (TGM), while the sum of all airborne species is indicated as total Hg (THg) [9,10]. The conversion among different forms is a dynamic process [11,12]: the knowledge of these processes is fundamental to understand the Hg biogeochemical cycle and to define the effects on human health. Among gaseous species, GEM is the most volatile and the predominant atmospheric form (typically >95% [13]), characterized by high stability and an elevated (6 months to 1 year) residence time in the atmosphere [14] due to its comparative chemical inertness. Thanks to these features, GEM can be transported up to thousands of kilometers from the source [15]. RGM mainly refers to divalent inorganic species ($HgCl_2$, HgO, $HgSO_4$) and to methylated forms (methyl- and dimethyl-Hg). Monovalent Hg compounds rarely occur in the atmosphere because they are unstable and rapidly oxidized to the stable state (i.e., $Hg^{2+}$ [16]). Reactive gaseous Hg consists of Hg compounds with sufficiently high vapor pressure to exist in the gas phase [17]: these forms are highly water-soluble, so their atmospheric residence time is sensibly lower than GEM [18]. PBM only has an operational definition, being defined by the pore size of the filter used for sampling [19]. PBM consists of both stable condensed Hg and gaseous Hg forms adsorbed on atmospheric particulate matter [20].

Mercury sources are both natural and anthropogenic. Mercury is naturally emitted in the atmosphere by geothermal activity, weathering of superficial rocks and soil, volcanic activity, but is also re-emitted from vegetation surface, from ocean and water bodies, or through wildfires [15,21]. Anthropogenic Hg sources are mainly linked to fossil fuel combustion, waste incineration, industrial processes like cement and building production, metal smelting, refining, and manufacturing [3,22]. Metallic Hg is still used in a variety of households and in industrial and medical products, while in artisanal and small-scale mining $Hg^0$ is used to extract gold from ore as an amalgam [21,23].

Humans are exposed to elemental Hg mainly by inhalation through the lungs, where $Hg^0$ is rapidly and almost completely adsorbed (ca. 80%) and readily distributed throughout the whole body [24]. The high diffusion of $Hg^0$ is due to the uncharged monoatomic form that ensures lipid solubilization: $Hg^0$ can rapidly cross the blood–brain or blood–placenta barrier, and it accumulates in the brain or in the fetus once oxidized [25]. In the gastrointestinal tract, GEM is instead poorly adsorbed (<0.01%) and almost completely excreted in the feces [25,26]. Dermal exposure to $Hg^0$ may also result from contaminated air [27], especially in work settings, even if the absorption is limited (ca. 2–3% of the inhaled dose) [11,25,27]. The toxicity to $Hg^0$ is mainly manifested in the central nervous system and in the kidney. Tremor is assumed as the early and the principal neurological effect of $Hg^0$ chronic

exposure, related with several secondary effects like memory deficits and motor system disturbances [28]. Kidney damage due to the toxic effect of $Hg^0$ is mainly manifested as proteinuria and various nephrotic syndromes [25]. A wide range of toxic effects have also been observed in cardiovascular (palpitations, increase in the blood pressure) and gastrointestinal (stomatitis, pain, nausea) systems [24]. An occupational exposure to $Hg^0$ vapor has been demonstrated to be an infertility factor for women [29].

The inhalation exposure to inorganic $Hg^{2+}$ compounds results in a systemic absorption in several body regions (head, kidneys, pelvis, and legs) thanks to the high affinity of $Hg^{2+}$ with blood thiolate anion and intracellular thiols (glutathione, glycinyl-cysteine); $Hg^{2+}$ compounds have also been found in human cord blood, placenta, and milk, indicating Hg transfer to the fetus and infants [30]. Differently, inorganic $Hg^{2+}$ compounds are only partially adsorbed by the gastrointestinal tract after ingestion (7–15% of doses), and they are almost eliminated by feces [31]. There is no evidence of mutagenic and carcinogenic effects resulting from inorganic Hg exposure in humans: only a few studies have reported adverse effects in animals, though these data have been considered inadequate [32].

## 3. Limit Values: Definition, Classification and Derivation Methods

The threshold limit value was firstly defined in the middle of the 1950s during the ACGIH (American Conference of Government Industrial Hygienists) meeting as "the concentration of a substance that should cause no significant injury to the health of the large majority of persons exposed daily" [33]. This definition has been subsequently modified to "airborne concentrations of chemical substances and represent conditions under which it is believed that nearly all workers may be repeatedly exposed, day after day, over a working lifetime, without adverse health effects" [34]. Based on these definitions, TLVs were developed to ensure people's health during occupational exposure; nevertheless, in the last years, several organizations (Agency for Toxic Substances and Disease Registry—ATSDR, Unite States Environmental Protection Agency—EPA, California Environmental Protection Agency—CalEPA and the World Health Organization—WHO) have also suggested limit values for residential and non-working settings.

TLVs are generally expressed in $mg/m^3$, so the only form considered for workers exposure is through inhalation, despite the contribution of other exposure pathways, like oral ingestion and dermal absorption, that not always could be considered as negligible for the toxic effect of a chemical [35]. TLVs usually refer to an exposure time. Occupational (i.e., work) exposure concentration limits are often expressed as time-weighted average (TWA), i.e., a concentration averaged over a specific time period, such as 8 h day and 40 h week [34,36,37] or 10 h working day [38] (see Table 1). For example, the Occupational Safety and Health Administration (OSHA), the regulatory agency of the United State (US) Department of Labor, refers to an 8 h TWA with the name of PEL (permissible exposure limit) as "the maximum permitted 8-h time-weighted average concentration of an airborne contaminant" [39]. The National Institute for Occupational Safety and Health (NIOSH), the US federal agency that is appointed to safeguard the workers safety and health, proposes a 10 h TWA named as REL (recommended exposure limit), i.e., "a time-weighted average concentration for up to a 10-h workday during a 40-h workweek to protect workers from hazardous substances and conditions in the workplace" [38]. US EPA proposed acute exposure guideline levels (AEGLs, see definition in Table 1) even for shorter time-intervals, from 10 min to 8 h (US EPA 2010) [40].

TWA exposures may also refer to concentration that must not be exceeded to avoid the potential health risks of chronic effects over long time exposure; this is, for example, the case of the EPA reference concentration (RfC, [41]) and of ATSDR action levels [42]; for their definition, see Table 1. In the definition of reference exposure levels (RELs, see Table 1), recently proposed by CalEPA [43] the health effects of the exposure to a contaminant may become more explicit with respect to TLV-TWA thresholds. RELs, for example, refer to a "[...] concentration at or below which adverse health effects are not likely to occur in the general human population"; thus, they represent threshold that should not be exceeded.

REL may be designed for both chronic and occupational exposures (see definitions in Table 1). Interestingly, in 2014, CalEPA also defined an acute REL as the level at which "intermittent one-hour exposures are not expected to result in adverse health effects" [43].

**Table 1.** Definitions of the TLV found in the literature.

| TLV Acronym/Name (Extended Name) | Definition | TLV Type/Note | Bibliography |
|---|---|---|---|
| Action level | *"Indoor air concentration of mercury vapor that should prompt public health and environmental officials to consider implementing response actions"* | No TWA. Action level could refer to residential or workplace settings. | [42] |
| AEGLs (Acute Exposure Guideline Levels) | *"Threshold exposure limits for the general public, applicable to emergency exposure periods ranging from 10 min to 8 h."* | Three levels (AEGL-1, AEGL-2, AEGL-3) are developed for each of five exposure periods (10 and 30 min, 1 h, 4 h, and 8 h) and are distinguished by varying degrees of severity of toxic effects. | [40] |
| Ceiling limit | *"Ceiling concentrations that must not be exceeded during any part of the workday"* | If instantaneous monitoring is not feasible, the ceiling must be assessed as a 15-min TWA exposure. Regulatory limit. | [44] |
| IDLH (Immediately dangerous to life or health) | *"A condition that pose an immediate threat to life or health, or conditions that pose an immediate threat of severe exposure to contaminants (…) which are likely to have adverse cumulative or delayed effects on health"* | Based on the effects that might occur as a consequence of a 30-min exposure | [45] |
| IOELV (Indicative occupational exposure limits) | *"Health-based, non-binding values, derived from the most recent scientific data available and taking into account the availability of measurement techniques"* | IOELVs are established by the European Commission, assisted by the Scientific Committee for Occupational Exposure Limits to Chemical Agents (SCOEL). | [46] |
| LOAEL (Lowest Observable Effect Levels) | *"The lowest exposure (or dose) level of a chemical at which there are statistically or biologically significant increases in frequency or severity of adverse effects between the exposed population and its appropriate control group"* | | [41] |
| MRL (Minimal Risk Level) | *"An estimate of daily human exposure to a substance that is likely to be without an appreciable risk of adverse effects (noncarcinogenic) over a specified duration of exposure"* | | [30] |
| NOAEL (No Observed Adverse Effect Level) | *"The highest exposure (or dose) level of a chemical at which there are no statistically or biologically significant increases in frequency or severity of adverse effects seen between the exposed population and its appropriate control"* | | [41] |
| PEL (Permissible Exposure Limit) | *"The maximum permitted 8-h time-weighted average concentration of an airborne contaminant"* | Regulatory limit. | [39] |
| REL (Reference Exposure Level) | *"A concentration at or below which adverse health effects are not likely to occur in the general human population"* | REL could refer to chronic expositions or TWA. | [43] |
| REL (Recommended Exposure Limit) | *"A time-weighted average concentration for up to a 10-h workday during a 40-h workweek to protect workers from hazardous substances and conditions in the workplace"* | Refer to a TWA exposition. | [47] |

**Table 1.** *Cont.*

| TLV Acronym/Name (Extended Name) | Definition | TLV Type/Note | Bibliography |
|---|---|---|---|
| RfC (Reference Concentration) | *"An estimate (with uncertainty spanning perhaps an order of magnitude) of a daily inhalation exposure of the human population (including sensitive subgroups) that is likely to be without an appreciable risk of deleterious effects during a lifetime"* | | [41] |
| TCL (Lowest Toxic Concentration) | *"The lowest concentrations known to cause any level of harm to humans"* | | [42] |

All the threshold limit value derivations basically started from epidemiological data found in the literature, namely, published peer-reviewed studies examined by committees composed by members from academia, government, or industrial settings, which critically evaluate the toxicological data coming from animal experiments or, preferentially, epidemiological data on humans [48]. Committee members consider several features of the toxic substance, like physical and chemical properties, the toxic/pharmacokinetics, the potential pathways of exposure, and its ability to cross the biological membranes [49]. Afterwards, the concentration of the chemical compound in biomarkers (urine, blood, hair, nails) of workers or animals allows for an estimation of the concentration of that chemical in air and a NOAEL or a LOAEL (see definitions in Table 1) at which no adverse/adverse effects have been observed in exposed people/animals. To obtain the TLV, the highest concentration of a toxic substance at which no adverse health effects are observed (NOAEL) or the lowest concentration of a toxic substance at which adverse health effects are observed (LOAEL) is frequently divided by an uncertainty factor (UF [30]), i.e., a mathematical adjustment applied to account for variations in people's sensitivity, the differences of the toxicity data between animals and humans, the uncertainty in using occupational data for another type of exposure, and for the use of NOAEL or LOAEL data. For example, as we will see later for Hg TLVs, the minimum risk level (MRL [30]) suggested by ATSDR or the RfC proposed by US EPA [41] are both calculated starting from a LOAEL that is then divided by an UF.

However, not all the TLVs are calculated employing an UF: as we will see later, this is the case of the occupational TLV for Hg established by the European Union and adopted by all the EU member states [46].

**4. Threshold Limit Values for Hg**

The literature revision allowed to detect twenty-three Hg TLVs: six of these refer to a chronic exposure and seventeen are calculated for a TWA exposition (Table 2). The detailed description of the derivation method of each TLV is reported in the Summary S1 of the Supplementary Materials (SM).

**Table 2.** Hg TLVs for residentials and workplace environments. TVLs with a regulatory value are underlined.

| | $\mu g/m^3$ | Bibliography |
|---|---|---|
| *chronic exposure* | | |
| CalEPA—chronic REL for $Hg^0$ | 0.03 | [43] |
| ATSDR—MRL for chronic-duration inhalation | 0.3 | [30] |
| EPA—RfC | 0.3 | [41] |
| WHO—Average annual Hg concentration guideline | 1 | [50] |
| ATSDR—Action level for residential settings (normal occupancy) | 1 | [42] |
| ATSDR—Action level for residential settings (residents' evacuation/relocation) | 10 | [42] |

**Table 2.** *Cont.*

| | μg/m$^3$ | Bibliography |
|---|---|---|
| *TWA exposure* | | |
| CalEPA—8 h REL for Hg$^0$ | 0.06 | [43] |
| CalEPA—acute REL for Hg$^0$ | 0.6 | [43] |
| ATSDR—Action level for workplaces not covered by 29 CFR 1910 Subpart Z | 3-4 | [42] |
| European Union 8-h TWA IOELV (Directive 2009/161/EU) | 20 | [32] |
| ATSDR—Action level for workplaces covered by 29 CFR 1910 Subpart Z | 25 | [42] |
| ACGIH—TLV (8-h TWA for inorganic Hg forms including metallic Hg) | 25 | [51] |
| Cal/OSHA—PEL (8-h TWA for mercury, metallic and inorganic compounds as Hg) | 25 | [44] |
| WHO—Recommended occupational exposure (TWA) | 25 | [52] |
| OSHA PEL (8-h TWA) and NIOSH—REL (10-h TWA) | 50 | [53]; [47] |
| NIOSH and OSHA Ceiling limit for metallic and inorganic Hg | 100 | [44]; [47] |
| EPA AEGL-2—AEGL-3 for 8 h exposure | 330–2200 | [40] |
| WHO—Recommended occupational short-term exposure | 500 | [52] |
| EPA AEGL-2—AEGL-3 for 4 h exposure | 670–2200 | [40] |
| EPA AEGL-2—AEGL-3 for 60 min exposure | 1700–8900 | [40] |
| NIOSH IDLH | 10,000 | [45] |
| EPA AEGL-2—AEGL-3 for 30 min exposure | 2100–11,000 | [40] |
| EPA AEGL-2—AEGL-3 for 10 min exposure | 3100–16,000 | [40] |

### 4.1. Chronic Hg TLVs

The Hg TLVs referred to a chronic exposition range from the lowest value of 0.03 μg/m$^3$ set by CalEPA (REL for Hg$^0$; [43]), to the maximum value of 10 μg/m$^3$ released by ATSDR [42] as the action level for residents' evacuation/relocation in residential settings (see Table 1 and Summary S1 in the SM for the complete definition). Several chronic Hg TLVs are calculated starting from various occupational studies [54–58], whereas the derivation method employed by each agency is not the same (see Summary S1 in the SM). This is, for example, the case of the CalEPA REL (0.03 μg/m$^3$) and the EPA Rfc (0.3 μg/m$^3$). Both limits ultimately refer to the same issue and aim (see Table 1), that is, avoid adverse health effects to the exposed population. Both the agencies start from a LOAEL of 25 μg/m$^3$, adjusted to a continuous exposure duration (LOAEL-ADJ) of 9 μg/m$^3$, which is then divided by a UF factor of 300 and 30 by CalEPA and EPA, respectively [43,59]. Differently, ATSDR calculates its MRL (0.3 μg/m$^3$, [30]), converting the Hg concentrations of workers' biomarkers (i.e., urine) reported in several occupational exposure studies [54,60–66] to an equivalent Hg$^0$ air concentration (2.84 μg/m$^3$): this value is then divided by a UF of 10 resulting in about 0.3 μg/m$^3$ as a chronic Hg TLV.

WHO proposes an average annual concentration of 1 μg/m$^3$ for Hg vapor (the chemical form is not specified) as a limit for continuative (i.e., chronic) exposure [50]. This value is calculated starting from a LOAEL which varies over a wide concentration range (from 15 to 30 μg/m$^3$), derived from an occupational study [67] and from a previous WHO report [68]. The LOAEL is then divided by a cumulative UF of 20, leading to a Hg TLV of 1 μg/m$^3$ through a procedure whose single steps are not entirely clear.

The same TLV (1 μg/m$^3$) is also proposed by ATSDR [42] as the action level for normal occupancy in residential settings, while, as reported above, the action level for isolation/evacuation of people from residential settings is set by ATSDR at 10 μg/m$^3$, following the study by Ngim et al. [58].

### 4.2. TWA Hg TLVs

The Hg TLV TWAs are typically related to an occupational exposure that occurs with a specific duration and frequency. The lowest value is proposed by CalEPA [43], who suggests an 8 h REL of 0.06 μg/m$^3$.

CalEPA, in addition to this latter limit, also releases an acute REL (0.6 μg/m$^3$, [43]). The 8 h REL of 0.06 μg/m$^3$ is substantially derived as the chronic REL (see Section 4.1), starting from a LOAEL of 18 μg/m$^3$, assuming that the ventilation rate for humans during

a working day is half (10 m$^3$/day) that of the whole day (20 m$^3$/day). The acute REL (0.6 µg/m$^3$) refers instead to a one-hour intermittent exposure and is calculated from the study of Danielsson et al. [69] carried out on a group of rats exposed to Hg$^0$ vapors.

In addition to those for chronic exposition (see Section 4.1), ATSDR [42] also proposes a TLV in the range of 3–4 µg/m$^3$ as the action level specific for those workplaces not covered by the Subpart-Z of the 29th Title of the Code of Federal Regulation [37], i.e., for those workplaces where an exposure to a toxic compound is not expected (see Summary S1 in SM for further details).

The European Union (EU) established an occupational 8 h TWA of 20 µg/m$^3$ for "Hg and divalent inorganic compounds of Hg, including Hg oxide and Hg chloride (measured as Hg)" with the Directive 2009/161/EU [46]: this directive, transposed by all the EU member states, reports a list of indicative occupational exposure limits (IOELVs) for several chemical compounds. The EU Hg 8 h TWA is indicated by the Scientific Committee for Occupational Exposure Limits to Chemical Agents (SCOEL): the suggested TLV for Hg$^0$ (20 µg/m$^3$, [32]) is derived by the Hg levels found in workers' biomarkers with central nervous system toxicity effects [56,70,71].

Several agencies (ATSDR, ACGIH, Cal/OSHA, WHO) concur to recommend an 8 h TWA of 25 µg/m$^3$ as occupational TLV, considering both metallic Hg and other forms of inorganic Hg. Specifically, this value is suggested by (i) ATSDR, as the action level for workplaces covered by the 29 CFR 1910 Subpart-Z [42], (ii) by ACGIH [51], (iii) by WHO, as a recommended occupational exposure [52], (iv) by Cal/OSHA, as an 8 h PEL [53]; this last limit is transposed into the California Code of Regulations.

The other Hg TLVs legally transposed as federal US laws are the 8 h TWA of 50 µg/m$^3$ and the OSHA ceiling limit (see definition in Table 1) of 100 µg/m$^3$ [44]. These values refer to Hg$^0$ and are derived from several occupational studies [72–74]; all the authors find that adverse effects to vapor Hg exposure occur at concentrations ranging between 50 and 100 µg/m$^3$, which are assumed by OSHA as an 8 h TWA and a ceiling limit, respectively. The same ceiling value is reported by NIOSH as a ceiling REL [47].

A similar definition of the OSHA acceptable ceiling concentration and NIOSH ceiling REL is the recommended occupational short-term exposure (see Summary S1 in SM) of 500 µg/m$^3$, released by WHO [52].

The NIOSH IDLH for all inorganic Hg compounds is set at 10,000 µg/m$^3$ [45]. As reported by NIOSH [45], IDLHs were based on the effects that might occur as a consequence of a 30 min exposure, therefore an exposure to concentration of 10,000 µg/m$^3$ must be avoided even for a very short time span.

Finally, we report the AEGLs released by EPA [40] for time intervals ranging between 8 h and 10 min exposure, divided according to the severity of the toxic effects (for specifications, see Summary S1 in SM). EPA AEGLs for 8 h exposure to Hg$^0$ vary between 330 µg/m$^3$ and 2200 µg/m$^3$ for AEGL-2 and AEGL-3, respectively; for a 10 min exposure, the values rise to 3100 µg/m$^3$—16,000 µg/m$^3$ [40]; this last is the highest TLV and the shortest TWA for Hg found in the current technical literature.

## 5. Discussion

The large amount of gaseous Hg TLVs found in the literature gives rise to the two most important themes for discussion: (i) the problematic division between chronic and TWA (i.e., occupational) TLVs, and (ii) the different derivation methodologies through which the TLVs are proposed and, therefore, the different TLV values referred to almost the same conditions of application. The question of airborne Hg is of central importance if we consider that gaseous Hg is a ubiquitous pollutant; in fact, different from the Hg pollution of water or food, that could diversely affect people depending on their lifestyle and diets, atmospheric Hg$^0$ pollution concerns all mankind, and it is also the Hg fraction most bioavailable, and therefore the most dangerous.

The time of exposure to a chemical compound is probably the foremost parameter considered to define TLVs. Environmental agencies mainly divide TLVs between residential

(i.e., chronic), occupational (i.e., intermediate), and acute exposure. The first question results from this division and from the differences that could be found between the TLV definitions. The definition of chronic exposure is almost similar for the different environmental agencies. For example, EPA defines chronic exposure as a "repeated exposure by the oral, dermal, or inhalation route for more than approximately 10% of the life span in humans (. . .)" [75]. Similarly, ATSDR in the MRL explanation defines the chronic exposure as the "contact with a substance that occurs over a long time (more than 1 year)" [30]. If we compare these definitions with a TWA limit definition, it is difficult to find substantial differences between an occupational and a chronic exposure. For example, the occupational limit released by SCOEL [32] and reported in the Directive 2009/161/EU refers to "repeated exposures (mainly 8-h per day, 5 days per week) over a working lifetime (up to 45 years)"; this time interval corresponds to more than 10% of the mean lifetime of most people. Moreover, the difference between a chronic and a TWA exposure seems to be even more inadequate if we specifically consider Hg and its slow elimination rates from the human body, as suggested by epidemiological data. Sällsten et al. [76] found a urinary Hg elimination half-life of 55 days among workers occupationally exposed for several years to Hg vapor; Hursh et al. [77] observed that, after an exposure of a few minutes to high concentrations (about 90 $\mu g/m^3$), the half-life elimination of Hg was 58 days from the whole body, while from the kidney, it was 64 days. These elimination rates are ever lower for people who were experimentally subjected to atmospheres with high GEMs, especially for women [78]. In this regard, the difficulty to find a substantial difference between an occupational and a residential (i.e., chronic) exposure led several authors [79,80] to lowering the existing $Hg^0$ limits. The issue between residential (i.e., chronic) and occupational exposures is deeply discussed by CalEPA [43]; not surprising, this agency proposes the lowest Hg TWA (8 h REL of 0.06 $\mu g/m^3$) found in the literature. Furthermore, it is notable that anomalous Hg vapor concentrations, mainly attributable to accidents with products containing Hg or to Hg release from unknown sources [81], could occur even at home. For example, Carpi and Chen [82] found $Hg^0$ concentrations in residential and business dwellings that exceeded EPA RfC (0.3 $\mu g/m^3$), while Li et al. [83] recorded even greater TGM values in several residential locations of Chongqing (China). These data suggest that a high background Hg pollution may be present even in residential settings, so TLVs for workplaces should consider the cumulative impact of both pollution sources (i.e., occupational and residential) on people's health, in order to develop correct safeguard guidelines.

An additional key point that should be discussed refers to the uncertainty factors (UFs) used to derive TLVs. This approach is not adopted by all the environmental agencies, or the application criteria are not always explained. Based on our research and the data reported in the Section 3, UFs are employed in TLV calculations by CalEPA, ATSDR, EPA, and WHO; however, the same values are not always adopted, although they are applied to the same procedure. For example, one of the UFs that is most used is the uncertainty factor to convert a LOAEL to a NOAEL, and hence to derive several Hg TLVs [43,50]. This UF is usually required because NOAEL is the preferable point of departure to calculate the TLV; if the NOAEL is not identified in the epidemiological studies, it is estimated dividing the LOAEL by an appropriate UF [84,85]. Despite its importance, the UF is not always employed with a constant value to calculate TLVs referring to the same exposition time. This is, for example, the case of chronic $Hg^0$ TLVs released by CalEPA (0.03 $\mu g/m^3$) and WHO (1 $\mu g/m^3$) that, starting from an almost identical LOAEL (see Summary S1), employed a UF of 10 and 2, respectively [43,50]. Similarly, in accounting for the variations in people's sensitivity to a toxic compound, the UF is not always considered with the same value. This is the case of chronic exposure to $Hg^0$, where ATSDR for the MRL calculation and WHO employed a UF of 10 [30,50], while CalEPA employs a UF of 30 [41]; the final resulting TLVs are thus quite different (Table 2). In addition, the selection of the UFs to be used in the TLV calculation is not uniform among the agencies. This is the case of CalEPA and EPA, which respectively use a UF factor of 300 and 30 [43,59] to calculate a chronic Hg TLVs. Thus, in this case, there are two different limits enforced in the same federation (USA), depending upon the state.

The freedom of the agencies in employing different UFs is probably based on subjective judgment among agency members considering the entire database of toxicologic effects for the different chemicals [86]. Despite this, for the same toxic compound, the UFs used should be the same, but probably for Hg, the great uncertainty to evaluate epidemiological evidence [87] causes a lack of uniformity in the adopted UF values. It is notable that, when specific chemical data are not available, a typical UF value of 10 is often employed for people's variability to calculate TLVs in residential settings [88]. Differently, in occupational studies, a UF of less than 10 is often employed, due to the more homogeneous composition of workers with respect to the general population [89]. However, this approach seems to not be so accurate; despite young, sick, or old people not forming an occupationally exposed population, workers may be rather heterogeneous; asthmatics, atopic people, pregnant women, and other susceptible categories can be included, as already noticed by Dankovic et al. [90]. It is also worth highlighting that several agencies which proposed Hg TLVs for TWA exposure (NIOSH, OSHA, and SCOEL; Table 2) did not employ UFs for TLV calculation. Based on their documentation [33,44,47], these agencies obtain TLVs directly from NOAEL, without employing UFs [44,47], or from biological values (Hg in urine or blood) directly converted to an atmospheric reference concentration [32]. The question is not of secondary importance based on the influence of these agencies on TLV release; NIOSH and OSHA jointly account for the USA Code of Federal Regulation, while SCOEL is the occupational exposure committee of the European Commission.

Moreover, the different exposure pathways (i.e., inhalation, ingestion, dermal absorption) of potential risk for human health are not always considered, despite people, both in residential and working settings, being exposed to a combination of Hg compounds in its different valence states that could be converted from one form to another via several chemical reactions [30,91]. For example, skin notation indicates the possibility that Hg vapor could be absorbed through the skin; this notation is reported in TLV dispositions by several environmental agencies [30,38,44,51,53]. In the 29 CFR 1910 act, OSHA underlines the necessity to add this skin notation; moreover, it suggests how to properly handle clothes that could be contaminated due to a continued exposure to airborne Hg compounds [44]. The skin notation is instead not reported by other agencies, such as the SCOEL; in its report on Hg [32], the agency states that, although "a small amount of skin absorption occurs on exposure to mercury vapor (. . .) the potential contribution to systemic body burden seems to be insufficient to merit application of the skin notation". The absence of the skin notation in the SCOEL factsheet is then also transposed in the Community Directive dealing with occupational exposure limit values [46]. Nevertheless, not all the Member states that transposed this Community Directive omit the skin notation; this is, for example, the case of the Italian law for health and safety on workplaces [92], where the skin notation for Hg is reported in association with the 8 h TWA (20 $\mu g/m^3$). Probably, the skin notation is not commonly reported in the TLV definition because it is not always recognized as a route of Hg exposure for humans. The question is strictly dealt with by the so-called MAK Commission, the Senate Commission for the Investigative of Health Hazards of Chemical Compounds in the Work Area, one of the commissions of the German Research Foundation (DFG). The MAK underlines the significant contribution of the Hg vapor skin absorption by pointing out that the 8 h TWA (20 $\mu g/m^3$) proposed by the Commission itself probably does not guarantee the prevention of an adverse effect on health [93].

Finally, it could be relevant to outline a timeline of Hg TLVs reported in the present paper, based on the year of first publication and on the values suggested/imposed (Figure 1). Based on this elaboration, we can observe a substantial lowering of the Hg TLVs proposed by the different agencies over time, with the lowest value among the chronic Hg TLVs that has been released only in the last years (i.e., EPA RfC [41]; CalEPA REL [43]; ATSDR MRL [30]). A substantial reduction can also be observed for occupational TLVs. This is the case of (i) the ACGIH TLV, firstly released at 50 $\mu g/m^3$ in 1994 and now set at 25 $\mu g/m^3$, (ii) the NIOSH IDLH, originally at 28,000 $\mu g/m^3$ and now suggested at 10,000 $\mu g/m^3$, and of (iii) the OSHA PEL, formerly enforced at 100 $\mu g/m^3$ and now estab-

lished at 50 μg/m$^3$ [44,45,51,94]. On the contrary, ATSDR MRL has been increased from its release in 1999 from 0.2 μg/m$^3$ to the more recent 0.3 μg/m$^3$ [30,95]. The constant reconsideration of the Hg TLVs reflects a lowering trend towards a more protective approach for people's health, whether the exposition to Hg occurs at home or at the workplace.

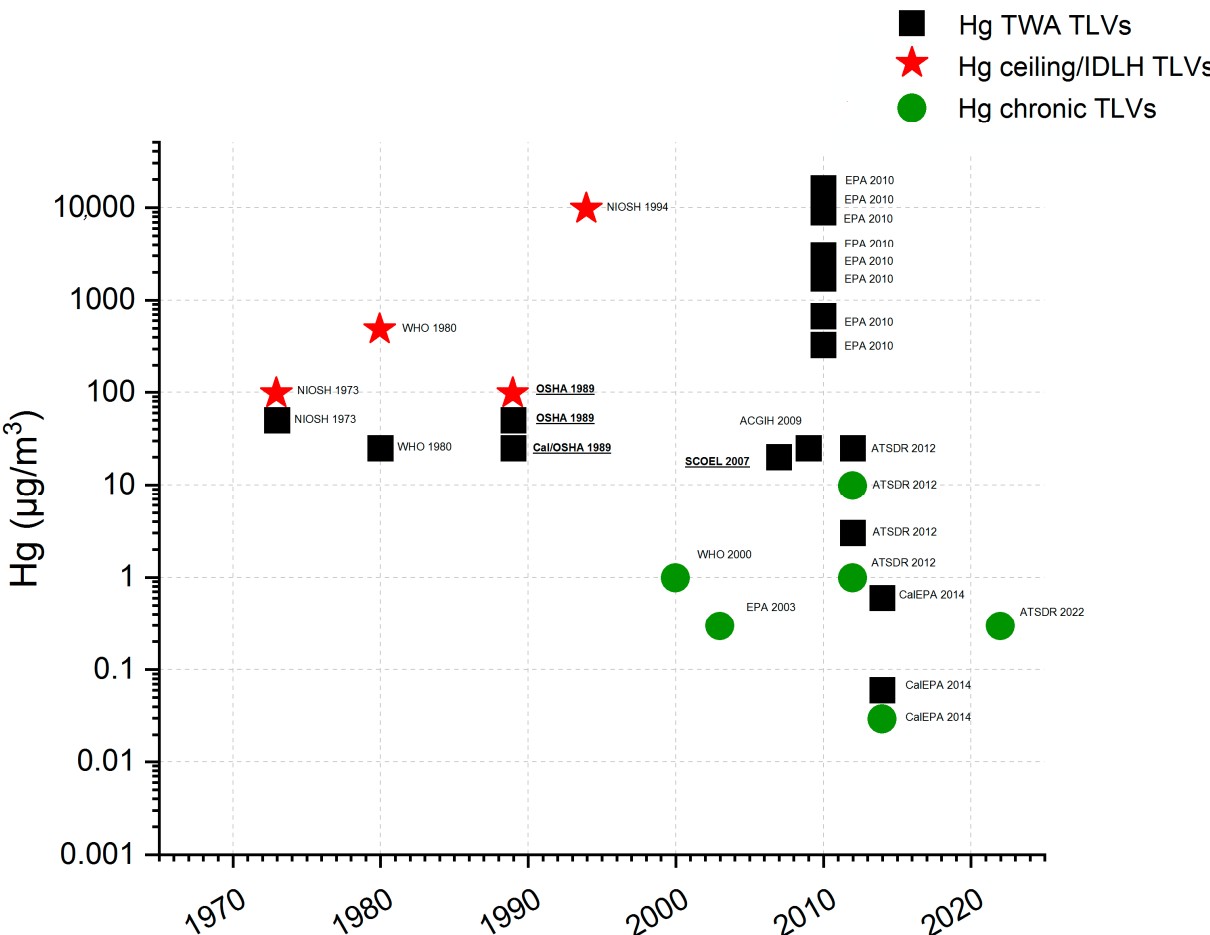

**Figure 1.** Timeline of Hg$^0$ TLVs for chronic (green dots), TWA (black squares), and ceiling/IDLH exposures in function of their values. TLVs with a regulatory value are underlined. (NIOSH 1973 [47]; WHO 1980 [52]; OSHA 1989 [44]; NIOSH 1994 [45]; WHO 2000 [50]; EPA 2003 [41]; SCOEL 2007 [32]; ACGIH 2009 [51]; EPA 2010 [40]; ATSDR 2012 [42]; CalEPA 2014 [43]; ATSDR 2022 [30]).

This reconsideration trend of Hg TLVs can also be understood on the basis of the last scientific evidence, which often report Hg concentrations of potential risk lower than most of the TLVs found in the literature. For example, in an epidemiological survey on people living in small-scale gold mining areas, Beate et al. [80] found a LOAEL of 3.5 μg/m$^3$.

ATSDR states that an exposure to a Hg$^0$ concentration greater than 10 μg/m$^3$ could be associated with human health effects [42]. This value (10 μg/m$^3$), assumed by ATSDR as the lowest toxic concentration level for humans (TCL), is also reported in other studies as the threshold value beyond which human urinary levels of Hg start to increase [96–98]. Similarly, Richardson et al. [79], based on Ngim et al. [58], refers to a LOAEL of 14 μg/m$^3$ as indicative for central nervous system effects related to an occupational Hg$^0$ exposure; this value is converted to 6 μg/m$^3$ LOAEL for a continuous (i.e., chronic) exposure. A concentration of 15 μg/m$^3$ is instead assumed by WHO [50] as the air Hg concentration which results in possible renal tubular effects, while 30 μg/m$^3$ is the supposed limit for objective tremors (see Summary S1 for details). These concentrations are sensibly lower than most of the legislative limits enforced for gaseous Hg, but close to the limit adopted by the EU for the 8 h TWA (20 μg/m$^3$), for example.

## 6. Conclusions

Airborne Hg pollution is a serious problem that received increasing attention during the last years, as demonstrated by the numerous scientific studies dedicated to this question. The problem has been addressed through the establishment of TLVs, that show a significant disagreement among the proposing entities, arising from the quite different approaches for establishing the limits. In recent years, a trend toward lowering the limits has been observed, reflecting a more conservative approach to human health. A key point is apparently the problematic division between chronic and TWA exposures. Moreover, several factors, mainly related to Hg exposure routes and the effects on human health, are not evenly considered. It seems therefore that a worldwide-coordinated effort (e.g., through UNEP) is needed to establish universally accepted criteria to define reliable and comprehensible TLVs.

**Supplementary Materials:** The following supporting information can be downloaded at: https://www.mdpi.com/article/10.3390/su16083142/s1, Summary S1: Threshold limit values for Hg.

**Author Contributions:** Conceptualization, F.C. and P.C.; methodology, F.C., P.C., P.L. and V.R.; software, F.C.; validation, P.C., P.L. and V.R.; formal analysis, F.C.; investigation, F.C.; resources, P.C. and V.R.; data curation, F.C.; writing—original draft preparation, F.C.; writing—review and editing, P.C., P.L. and V.R.; visualization, F.C., P.C., P.L. and V.R.; supervision, P.C., P.L. and V.R.; project administration, P.C. and V.R.; funding acquisition, P.C. and V.R. All authors have read and agreed to the published version of the manuscript.

**Funding:** This research was conducted without external funding. The University of Florence provided the access to bibliographic resources.

**Conflicts of Interest:** The authors declare no conflicts of interest.

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
