# Peer review of "Gaseous Mercury Limit Values: Definitions, Derivation, and the Issues Related to Their Application"

_sustainability, doi:10.3390/su16083142_

Round 1
Reviewer 1 Report
Comments and Suggestions for Authors
In their article, the authors explain the permissible limits of mercury in various countries and the associated impact on human health. The problem is the dilemma of using equivalent reference values depending on the authoritative organization endorsing these standards. However, the authors did not present their own position on this issue and do not propose values that they consider correct. My assessment of the article is rather positive. The graphical representations are particularly clear and the tabular data is accurate. The conclusions drawn are logical and coherent, and the cited literature maintains scientific rigor.
Author Response
Thanks to the reviewer for the suggestions. We point out that the purpose of our manuscript is a comprehensive overview of the actual state of art of the policy on mercury pollution. The focus of the work is not to propose or suggest limit values, but to underline the differences in their calculation, the critical key-points and some apparent contradiction in the TLVs release. However, the suggestion of the referee raises a key point and therefore in the Conclusions section (lines 423-425) we propose that a central coordination in TLVs release is probably a viable solution to establish universally accepted criteria for TLVs calculation and thus consistent values.
Reviewer 2 Report
Comments and Suggestions for Authors
The paper provides a comprehensive overview of mercury (Hg) pollution, its sources, forms, health effects, and threshold limit values (TLVs) proposed by various environmental agencies. While the paper provides valuable information on Hg pollution, there are several areas that need improvement.
(1) The paper discusses the increase in anthropogenic Hg emissions as a major contributor to Hg pollution. However, it lacks quantitative data on Hg emissions. It would be helpful to include studies related to Hg emission inventory in the "Mercury forms, sources, and health effects" section.
(2) The author mentions significant differences in the Hg TLVs proposed by different agencies. However, it does not discuss how these TLVs are implemented in real-world scenarios. It would be interesting to know which TLVs are followed by industries and workplaces in different countries to set safe exposure limits for workers.
(3) Given the wide range of TLVs, a meta-analysis approach might be beneficial. This would involve statistical analysis of TLVs from different sources to draw quantitative conclusions and identify patterns or trends. Such an analysis could add rigor and depth to the discussion of TLVs.
(4) The uncertainty factor (UF) is a critical parameter in TLV calculation but there is considerable variation in its values among agencies. Some agencies even neglect the UF in their calculations. It would be helpful if the author could provide some insight into this issue, discuss the reasons for the variations, and suggest a reasonable range of UF values.
Despite the above limitations, the paper provides a valuable overview of Hg pollution and its TLVs. Therefore, I recommend that the paper be accepted for publication after a minor revision.
Author Response
Thanks to the reviewer for considering our paper suitable for publication. Responses to individual requests are listed below.
(1): “The quantitative data about anthropogenic Hg emissions are in fact reported in the introduction section (see lines 36-37).”
(2): “We point out that the manuscript already reports the applicability of TLVs in the different workplaces and worldwide scenarios (see results paragraph). TLVs legally transposed by environmental laws are reported in the manuscript for the different countries (mainly for the United States and for the European Community). Industries and workplaces must comply with their respective state laws.”
(3): “The reviewer suggests an interesting approach to analyze the data found in literature. However, we decided to focus our research specifically on Hg limit values of the so called western world (more or less arbitrarily identified in EU, USA, Canada, Australia, and New Zealand). This specific limitation has been added in the text (lines 53-54). Because of the limited amount of data, we consider a statistical analysis not suitable. We thus decided to graphically represent the TLVs in function of the year of release (Figure 1): this elaboration does not correspond to a statistical analysis, but offers a representation of the reduction trend of the Hg limit values in the last years.”
(4): “Thanks to the reviewer for the suggestion. We report some insights on UFs employment and their variation among the different agencies (see lines 315-319; 323-325; 331-332). We do not think to go deeper, because the main goal of our paper is to offer an overview of the current scenario of mercury limit values, underlying the most critical key-points in their applicability and derivation. As suggested in the Conclusions section (lines 423-425) we propose that a central coordination is probably a viable solution to establish universally accepted criteria consistent calculations””
Reviewer 3 Report
Comments and Suggestions for Authors
I suggest to the authors to include AEGL values in the analysis. https://www.epa.gov/aegl/mercury-vapor-results-aegl-program
Author Response
Following the reviewer suggestion, the recommended reference has been included in the text.